# Structural and Functional Analysis of SHP Promoter and Its Transcriptional Response to FXR in Zn-Induced Changes to Lipid Metabolism

**DOI:** 10.3390/ijms23126523

**Published:** 2022-06-10

**Authors:** Han Gao, Xing Fan, Qi-Chun Wu, Chuan Chen, Fei Xiao, Kun Wu

**Affiliations:** 1College of Marine Sciences, South China Agricultural University, Guangzhou 510642, China; 19865040582@163.com (H.G.); fx13006878436@163.com (X.F.); h15859679623@163.com (Q.-C.W.); cc18190392889@163.com (C.C.); xf15521300836@163.com (F.X.); 2University Joint Laboratory of Guangdong Province, Hong Kong and Macao Region on Marine Bioresource Conservation and Exploitation, Guangzhou 510642, China

**Keywords:** SHP, promoter analysis, lipid metabolism, zinc, fish

## Abstract

Zinc alleviates hepatic lipid deposition, but the transcriptional regulatory mechanisms are still unclear. In this study, we characterized the promoter of an SHP (short heterodimer partner) in a teleost *Pelteobagrus fulvidraco*. The binding sites of an FXR (farnesoid X receptor) were predicted by the SHP promoter, indicating that the FXR mediated its transcriptional activity. The site mutagenesis and the EMSA (electrophoretic mobility shift assay) found that the −375/−384 bp FXR site on the SHP promoter was the functional binding locus responsible for the Zn-induced transcriptional activation. A further study of yellow catfish hepatocytes suggested that the activation of the FXR/SHP is responsible for the effect of Zn on the decreasing lipid content. Thus, this study provides direct evidence of the interaction between the FXR and SHP promoter in fish, and accordingly elucidates the potential transcriptional mechanism by which Zn reduces hepatic lipid accumulation.

## 1. Introduction

Zinc (Zn) is an essential micronutrient for many important biochemical processes in vertebrates. In recent decades, extensive studies have focused on its essential roles or toxic effects on growth, survival, reproduction, histological changes, redox response and immunity [1,2,3]. Recently, there has been a growing body of studies that recognizes the importance of Zn on metabolic processes, especially on the lipid metabolism. Zn deficiency is associated with a higher risk of fatty liver disease and an impaired lipid profile [4]. Clinical trials in humans and animals demonstrate that Zn plays a pivotal protective role in regulating the lipid profile and metabolism [4,5,6]. In fish, Zn accumulation can reduce lipid deposition. For example, Wu et al. [7] found that waterborne Zn exposure for 56 days reduced hepatic lipid deposition in *S. hasta*. Both chronic waterborne and dietary Zn reduced the tissue lipid content of yellow catfish by inhibiting lipogenesis and stimulating lipolysis [8,9]. Hence, Zn is considered to have the potential to prevent excessive lipid accumulation and maintain lipid homeostasis. However, the underlying molecular mechanisms have not been fully characterized in fish.

Lipids are essential nutrients and play important roles in metabolic processes. In fish, the failure to maintain lipid homeostasis impedes a wide range of physiological processes [10]. In general, lipid homeostasis is characterized by the balance between lipogenesis and lipolysis, which are regulated by transcriptional factors and crucial genes. The farnesoid X receptor (FXR) and short heterodimer partner (SHP/NR0B2) are important nuclear receptor and transcription factors controlling lipid metabolism [11,12]. The SHP plays important roles in regulating the expression of genes, with roles in bile acid transport, lipid metabolism and gluconeogenesis [13]. In eukaryotic organisms, promoters have many cis-acting elements and can be bound by transcriptional factors, which activate the transcription initiation, thereby regulating gene expression. Studies in mice suggest that the FXR is an upstream regulator of the SHP and targets the SHP promoter [13]. In a teleost, extraordinarily little is currently known about the interaction between the FXR and SHP. Our recent study suggested that the FXR/SHP pathway may constitute a crucial link between the tissue Zn content and the prevention of lipid accumulation [12,14]. Thus, it is crucial to analyze the structure and function of the SHP promoter, which may decipher the mechanism of Zn-regulating target genes relevant to lipid metabolism and lead to new treatments for fatty liver diseases.

The yellow catfish (*Pelteobagrus fulvidraco*) is an omnivorous freshwater fish commonly farmed in China and other countries for the excellent fillets and high market value. Under intensive aquaculture, this fish species frequently exhibits excessive hepatic lipid deposition and severe fatty liver syndrome, which have adverse effects on its health [12]. Thus, it is especially important and meaningful to research ways to reduce lipid deposition and the underlying molecular mechanisms. Given the potential role of Zn on the FXR/SHP pathway and the role of the pathway on lipid deposition [12,14], we hypothesize that the SHP acts as a mediator in the Zn-induced effects on lipid metabolism. In the present study, we identified the SHP promoter region of yellow catfish and investigated its functional bind with the FXR under the Zn signal. Our study offers innovative insights into the mechanism of Zn-regulating SHP transcriptional activity and lipid deposition, which are important for the evaluation of Zn nutrition and the regulatory effects on lipid metabolism in vertebrates.

## 2. Results

### 2.1. Cloning and Sequence Analysis of the SHP Promoter

In the present study, the 1790 bp SHP promoter of a yellow catfish was successfully cloned (Figure 1A). On the SHP promoter region, we predicted several core promoter elements, such as a TATA-box (TBP) located from −81 bp to −90 bp and a CCAAT-box (NF-Y) located at −101 bp to −110 bp (Figure 1B). We also predicted a cluster of binding sites of several transcription factors on the SHP promoter, including the FXR, EERγ (estrogen receptor-related receptor γ), Sp1 (GC-box), STAT3 (signal transducer and activator of transcription 2), PPARα/RXR (peroxisome proliferator-activated receptor α/retinoid X receptor) and KLF4 (Kruppel-like factor 4) (Figure 1B).

### 2.2. The 5′-Deletion Assay of the SHP Promoter

In the present study, we randomly generated plasmids of a distinct size and selected four appropriate plasmids to perform the 5′-deletion assay. The sequence deletion from −1790 bp to −808 bp showed no significant effects on the luciferase activity, but the deletion from −808 to −346 bp reduced the luciferase activity significantly (Figure 2).

To investigate the response of the promoter to Zn, HepG2 cells were incubated with 50 μM Zn^2+^ for 48 h and then the 5′ deletion assay was performed. Compared to the control, Zn significantly increased the luciferase activities of pGl3-1790/+39, pGl3-1252/+39 and pGl3-808/+39, but had no effect on pGl3-346/+39. In the Zn-treated group, the sequence deletion between −1790 bp and −808 bp of the SHP promoter showed no significant influences on luciferase activity; however, further deletion to −346 bp down-regulated the luciferase activity (Figure 3).

### 2.3. Site Mutation Analysis of FXR Binding Sites on the SHP Promoter

Based on the results of the 5′-deletion assay, we performed the site mutation using the pGl3-1790/+39 plasmid. The mutation of the −267/−276 FXR binding site (Mut-FXR-2) did not change the Zn-induced elevation of luciferase activity, suggesting that this site played no role in the SHP transcriptional response to Zn. In contrast, the mutation of the −375/−384 FXR binding site (Mut-FXR-1) decreased the Zn-induced luciferase activity significantly. Similarly, the co-mutation of the −267/−276 and −375/−384 binding sites (Mut-FXR-1 and 2) also down-regulated the Zn-induced increase in luciferase activity, indicating that the −375/−384 FXR binding site positively mediated the activity of the SHP promoter (Figure 4).

### 2.4. FXR Bind with the SHP Promoter

Based on the mutation assays and transcriptional analysis, we speculated that the FXR could bind with the sequence from −375 to 384 bp of the SHP promoter, thereby activating the transcription of the SHP and its downstream genes. Thus, we further used the EMSA (electrophoretic mobility shift assay) to detect the physical interaction between the transcription factor FXR and the specific binding site on the SHP promoter. As shown in Figure 5, when the FXR binding sequence was used as the probe, the 100-fold unlabeled binding sites (between −375 and −384 bp of the SHP promoter) competed for binding. In contrast, the 100-fold unlabeled Mut-FXR1 binding sites declined this competition significantly, indicating that this region could be bound by the FXR. Meanwhile, the increased brightness of bands under the Zn treatment reflected that Zn promoted the binding process between the FXR and this binding position on the SHP promoter.

### 2.5. Effect of Zn on Lipid Metabolism of Yellow Catfish Hepatocytes

Compared with the control, 50 μM Zn significantly increased the intracellular Zn concentration (Figure 6A), but decreased the contents of lipid droplets (Figure 6B) and TG (Figure 6C). We further detected mRNA levels of the FXR/SHP pathway and its downstream genes. Zn activated the FXR/SHP pathway because of the increased expression of the *fxr* and *shp* (Figure 6D). For genes involved in lipogenesis, Zn down-regulated the expression of *accα* and *fas*, but had no effect on *g6pd* and *6pgd*. Additionally, the expression of *cpt1α*, a key gene in fatty acid β-oxidation, was increased by Zn incubation. In general, Zn promoted the lipolytic processes but suppressed the lipogenic processes in the yellow catfish hepatocytes.

## 3. Discussion

Previous studies indicated that the FXR and SHP play important roles in regulating lipid-metabolism-related genes in yellow catfish [12,14], but direct evidence has not been explored. This study, for the first time, cloned and characterized the structure and function of the SHP promoter region from a yellow catfish and investigated its transcriptional response to the FXR in Zn-induced changes in lipid metabolism.

The first step for exploring the mechanism of transcriptional initiation is the identification of the core promoter, which is located most proximal to the start codon and contains the RNA polymerase binding sites [15]. In this study, one TATA-box, one CAAT-box (NF-Y) and one GC-box (Sp1) binding sites were found in the core SHP promoter region (Figure 1). The TATA-box and CAAT-box were commonly found at upstream sites close to the TSS and helped to dock the RNA polymerase transcriptional complex [16]. In fact, only approximately 5%–7% of the eukaryotic promoters had a TATA-box [15], and TATA-less promoters usually possessed Sp1 binding sites, which drive the basal transcription function [17]. In mammals, different splicing yields two SHP promoters and both of the two isoforms contain a TATA-box, CAAT-box and GC-box [18,19]. The similar arrangement of core promoter elements between the yellow catfish and mammals indicate that their SHP promoters have similar transcriptional initiation modes and conserved functions.

The identification of transcription factor binding sites (TFBSs) helps to decipher the regulatory mechanisms of genes. The present study showed that the luciferase activities of the SHP promoter underwent almost no changes when the sequence increased from −1790 bp to −808 bp, which may be attributable to the lack of key binding sites in this region. In contrast, the luciferase activity decreased when the sequence from −808 to −346 bp was deleted (Figure 2). A further investigation found that this region contained a cluster of TFBSs, such as the FXR, STAT3, ERRγ, PPARα/RXR and SREBP1 (Figure 1B). In mammals, these transcription factors have been identified to activate the SHP promoter and regulate SHP expression [13,20,21]. Thus, the putative transcription factors in the region from −808 to −346 bp are likely to be positive regulators of the SHP gene (Figure 1A). Notably, most of these transcription factors engage in lipid metabolism and obesity [22], suggesting that the SHP probably plays a role in the regulation of lipid metabolism. Indeed, multiple lines of evidence demonstrate that the SHP regulates several metabolic pathways by responding to other nuclear receptors and transcription factors [13]. Further, Zn incubation markedly increased the activity of the SHP promoter from −808 to −346 bp (Figure 3), indicating that Zn promotes SHP transcription, in agreement with a previous study in mouse and human cells [23]. Studies have suggested that Zn stimulates lipolysis and inhibits lipogenesis [7,8,9]. Our recent studies demonstrated the importance of the FXR/SHP pathway in regulating genes involved in lipid metabolism [12], especially under Zn action [14]. Considering that two putative FXR binding sites were positioned between −808 and −346 bp, this promoter sequence may be the pivotal region that responds to the zinc signal and regulates lipid metabolism in yellow catfish.

We next explored whether the FXR exerts regulatory action directly through these response elements. According to the putative results, two FXR binding sites were located on the region from −808 to −346 bp. However, a site mutation on the −375/−384 FXR binding site, but not on the −267/−276 binding site, reduced the Zn-induced increase in promoter activity (Figure 4). Information is extremely scarce about the direct interaction between the FXR and SHP in fish. In the present study, results of an EMSA further confirmed that the −375/−384 bp sequence was a functional binding site, and that Zn incubation promoted the FXR binding to this site because of the stronger band in the Zn-treated group (Figure 5). Similarly, Hoeke et al. [24] reported that the FXR regulated human SHP expression primarily via direct binding to key elements on the SHP promoter. The FXR is a ligand-modulated transcription factor and tightly controls bile acid synthesis and hepatic lipid metabolism through the induction of the SHP [25]. Therefore, the −375/−384 bp FXR site may play a vital role in responding to the Zn-induced up-regulation of SHP promoter activity.

Having determined the fact that the FXR stimulates SHP transcription by integrating with its promoter, we next explored their roles in the Zn-induced regulation of lipid metabolism using primary hepatocytes of yellow catfish. As shown in Figure 6, the increase in the intracellular Zn content reduced the deposition of lipid droplets and TG, like other studies [4,7,8,9]. Previous studies emphasize the importance of the FXR/SHP pathway in the regulation of lipid metabolism [12,14]. A high correlation between the FXR/SHP mRNA levels and the expressions of other genes involved in lipid metabolism was likewise observed in this study (Figure 6D). In general, the activation of the FXR/SHP pathway up-regulated the mRNA levels of lipolysis genes (*atgl* and *cpt1α*) and down-regulated the expression of lipogenesis genes (*accα* and *fas*), in agreement with the observations of lipid droplets and TG. Animal experiments showed that the SHP is an important target gene of the FXR in the regulation of lipid metabolism [26]. The SHP alleviates hepatic lipid deposition and obesity by acting as a transcriptional repressor of other lipogenic genes [13,27]. Considering the results of the site mutation assays and the EMSA, the FXR could directly bind with the SHP promoter, which could potentially mediate the regulation of lipid metabolism by Zn.

In conclusion, we identified and characterized the promoter region of the SHP gene from yellow catfish. In addition, we investigated the important roles of the FXR/SHP in Zn-induced changes in lipid metabolism. The present study provides the first direct evidence for the interaction between the FXR and SHP in fish, and accordingly elucidates the potential mechanism by which Zn reduces hepatic lipid accumulation.

## 4. Materials and Methods

### 4.1. Experimental Animals and Reagents

Yellow catfish were obtained from a local commercial farm (Guangzhou, China). HepG2 cell lines were obtained from the Cell Resource Center of our college. Dulbecco’s Modified Eagle Medium (DMEM) and fetal bovine serum (FBS) were purchased from Gibco (Thermo Fisher Scientific, Waltham, MA, USA). The experiments complied with the ethical guidelines of South China Agricultural University (SCAU), Guangzhou, China, for the use of experimental animals and cells.

### 4.2. Promoter Cloning and Plasmid Construction

The hiTAIL-PCR (high-efficiency thermal asymmetric interlaced PCR) method [28] was used to clone the promoter sequences, and the protocols followed our previous studies [29,30]. The specific primers were listed in the Appendix A. The luciferase reporter constructs were produced via the purified PCR product and pGl3-Basic vectors (Promega, Madison, WI, USA), and the ClonExpress II One Step Cloning Kit (Vazyme, Piscataway, NJ, USA) was used to ligate the products. Based on the distance from the TSS (transcription start sites), we named the plasmids as pGl3-1790/+48 of SHP vector. Then, using the pGl3-1790/+48 vector as a template, we generated pGl3-346/+48, pGl3-808/+48, pGl3-1252/+48 vectors with the Erase-a-Base system (Promega, Madison, WI, USA). The primer sequences for plasmid construction are shown in Appendix A.

### 4.3. Sequence Analysis and Activities Assays of Luciferase

The transcription factor binding sites (TFBS) were predicted by the JASPAR database (http://jaspar.genereg.net/ (accessed on 10 December 2020)) and MatInspector (http://www.genomatix.de/ (accessed on 10 December 2020)). The reference binding site sequences are listed in the Appendix A.

Plasmid transfections and activity assays of luciferase followed our recent publications [29,30]. Briefly, HepG2 cells were cultured in DMEM medium (10% FBS) in an incubator with 5% CO_2_ at 37 °C. Prior to the transient transfection, cells were seeded in a 24-well cell culture plate at a density of 1.2 × 10^5^ and cultured for 24 h to reach 70%–80% convergence. Plasmids were transiently transfected into HepG2 using Lipofectamine 2000 (Invitrogen, Carlsbad, CA, USA). The reporter plasmids were co-transfected with 35 ng pRL-TK as control. After 4 h, DMEM (10% FBS) or DMEM (10% FBS) + 50 μM Zn were used to replace the transfection medium. After 24 h incubation, we used Dual-Luciferase Reporter Assay System to measure the relative luciferase activity according to the manufacture’s manuals.

### 4.4. Site Mutation Assays of FXR Binding Sites on the SHP Promoter

Site mutation assays were performed to identify the FXR binding sites on the SHP promoter. The pGl3-1790/+48 vector was used as the template and the QuickChange II Site-Directed Mutagenesis Kit (Vazyme, Piscataway, NJ, USA) was used to conduct site-directed mutagenesis. The primers for mutagenesis are shown in Appendix A. The constructs were named Mut-FXR-1 and Mut-FXR-2. The pRL-TK and constructs were transfected into HepG2 according to the methods described in Section 4.3. After 24 h incubation, cells were harvested and the luciferase activities were detected.

### 4.5. Analysis of Functional Binding Sites of FXR on the SHP Promoter

An electrophoretic mobility shift assay (EMSA) was conducted to determine the functional binding site of FXR on the promoter region of SHP. Proteins for electrophoretic mobility shift assay (EMSA) were extracted from primary hepatocytes of yellow catfish. The EMSA protocol followed the instruction of the LightShift Chemiluminescent EMSA Kit (Invitrogen, Carlsbad, CA, USA) as in our previous studies [29,30]. Briefly, each oligonucleotide duplex of the FXR binding sites were incubated with 10 µg nuclear extracts. Before the biotin-labeled probe was added, each unlabeled probe was pre-incubated for 10 min. The biotin-labeled probe was added at room temperature and the reaction continued for 30 min. Then, they were detected via electrophoresis on 6% native polyacrylamide gels. This research performed competition analyses by using 100-fold unlabeled oligonucleotide duplex with or without the mutation. The oligonucleotide sequences of EMSA are shown in Appendix A.

### 4.6. Zn, LD and TG Contents in Hepatocytes and Gene Expression

We isolated hepatocytes from yellow catfish liver tissues according to our recent publications [12]. After 48 h incubation, intracellular Zn and LD were detected based on the protocols of our previous studies [12,31]. For detection of intracellular Zn^2+^ concentration, cells were incubated with 1 μM Newport Green DCF (Invitrogen, Carlsbad, CA, USA) for 30 min. For intracellular LD staining, cells were incubated with 5 mg/mL Bodipy (Invitrogen, Carlsbad, CA, USA) for 30 min. Fluorescence was imaged using laser scanning confocal microscopy (Leica, Wetzlar, Germany). We used the commercial assay kits (Nanjing Jian Cheng Bioengineering Institute, Nanjing, China) to analyze intracellular TG content. The mRNA levels of genes involved in lipid metabolism were examined by quantitative real-time PCR (Q-PCR). Primers are given in the Appendix A.

### 4.7. Statistical Analysis

SPSS 22.0 software (SPSS, Chicago, IL, USA) was used to conduct statistical analyses. Results are presented as mean ± standard errors of means (SEM). Before analysis, we used the Kolmogorov–Smirnov test to determine the normality of distribution of all data. The Student’s *t*-test was used to compare the differences between wild types and treated group. Significant levels were set at *p* < 0.05.

## Figures and Tables

**Figure 1 ijms-23-06523-f001:**
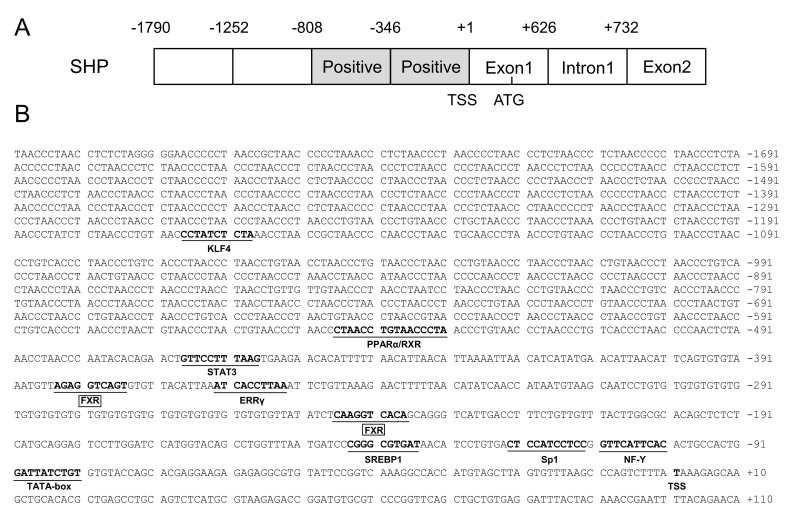
(**A**) The schematic diagram of SHP gene structure. The first nucleotide of 5′ cDNA of SHP was designated as +1. Positive: the region that positively regulated the promoter activity. TSS: transcription start site. ATG: translation initiation site. (**B**) Nucleotide sequence of yellow catfish SHP promoter. Numbers are relative to the transcription start site (+1). The putative transcription factor binding sites are underlined. The highlighted sequences show putative transcription factor binding sites of FXR.

**Figure 2 ijms-23-06523-f002:**
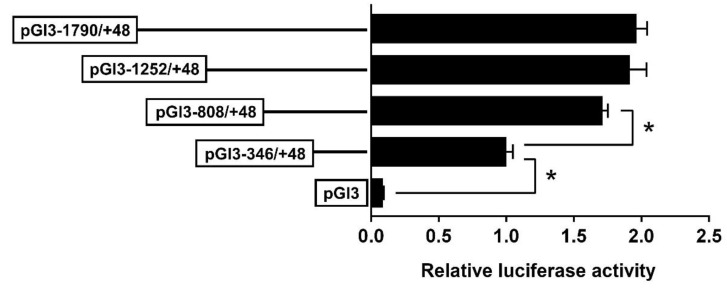
The 5′ unidirectional deletion assays of the SHP promoter region of yellow catfish. Values mean the ratio of activities of firefly to Renilla luciferase normalized to the control plasmid. Results are shown as mean ± SEM (*n* = 3). Asterisk (*) means significant differences between the two groups (*p* < 0.05).

**Figure 3 ijms-23-06523-f003:**
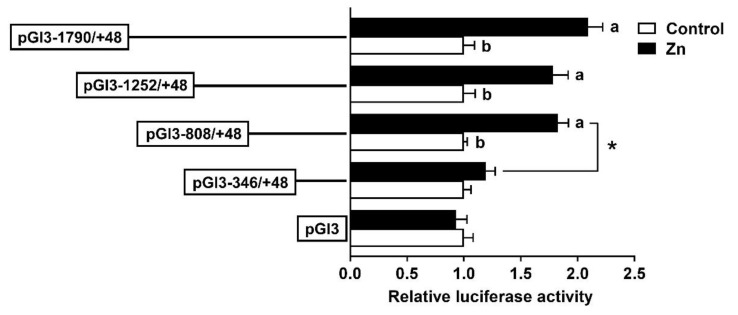
The 5′ unidirectional deletion assays for promoter regions of SHP after 50 μM Zn treatment. Values show the ratio of activities of firefly to Renilla luciferase normalized to the control. Results are presented as mean ± SEM (*n* = 3). Asterisk (*) indicates significant differences between different 5′ unidirectional deletion plasmids under the same treatment (*p* < 0.05). Different letters indicate significant differences between the different treatments in the same plasmid (*p* < 0.05).

**Figure 4 ijms-23-06523-f004:**
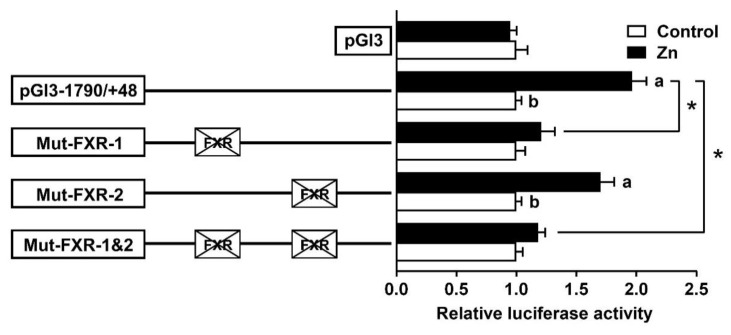
Assays of predicted FXR binding sites after site-directed mutagenesis. Values mean the ratio of activities of firefly to Renilla luciferase normalized to the control. Results are presented as mean ± SEM (*n* = 3). Asterisk (*) indicates significant differences between the different mutation plasmids under the same treatment (*p* < 0.05). Different letters indicate significant differences between different treatments in the same plasmid (*p* < 0.05).

**Figure 5 ijms-23-06523-f005:**
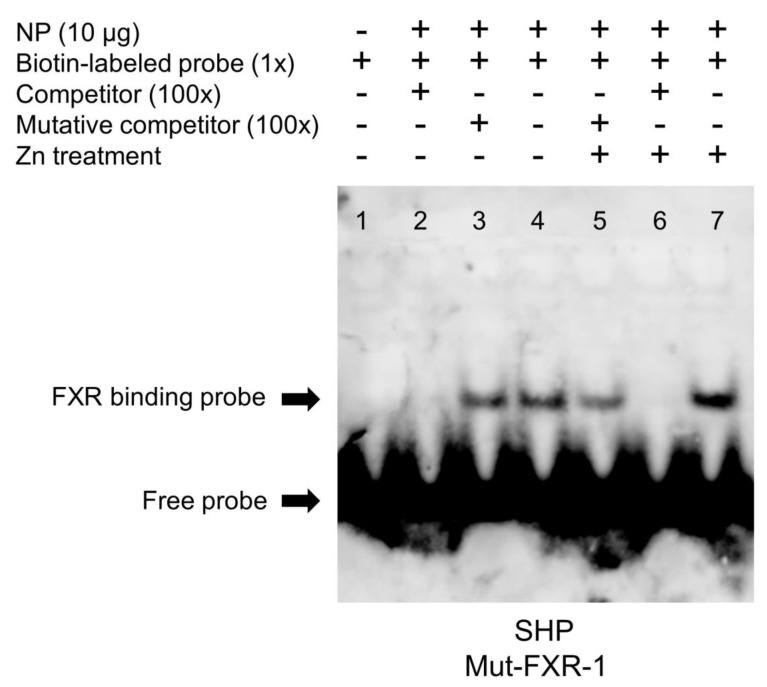
EMSA of predicted FXR binding sequence on yellow catfish SHP promoter (between −375 bp and −384 bp). NP, nuclear protein. The numbers 1–7 represent the 7 different lanes.

**Figure 6 ijms-23-06523-f006:**
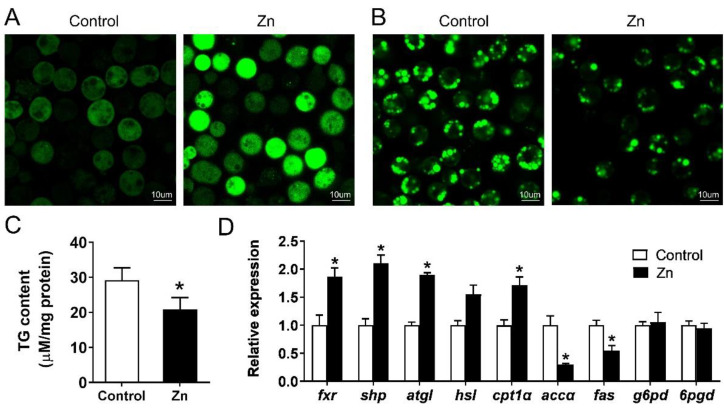
Effect of Zn incubation on cellular Zn concentration. (**A**): lipid droplet contents; (**B**): TG accumulation; (**C**): mRNA levels involved in lipid metabolism; (**D**): in hepatocytes of yellow catfish at 48 h. Results are presented as mean ± SEM (*n* = 3). Asterisk (*) indicates significant differences in different treatments (*p* < 0.05).

## Data Availability

Not applicable.

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
