# Peer review of "Structural and Functional Analysis of SHP Promoter and Its Transcriptional Response to FXR in Zn-Induced Changes to Lipid Metabolism"

_ijms, 2022, doi:10.3390/ijms23126523_

Round 1

Reviewer 1 Report

The authors have investigated and submitted a research article titled Structure and functional analysis of SHP promoter and its transcriptional response to FXRinZn-induced changes in lipid metabolism. There is a significant amount of novelty in this study that would help the current researchers who are working in this field. The current study focuses more on the journal aims. The experimental design and experimentation are fine. It may be more appropriate for publication in your journal after revision.

In this study, HepG2 cells were used instead of adipocytes since the main focus of the study was to investigate the role of Zn in modulating FXR-SHP, which plays a role in lipid metabolism.

The authors have described several genes that are associated with lipogenesis as well as lipolysis. I would like to know whether the authors checked PPAR expression in both experimental models as well as its downstream targets.

Please include the name of the medium used for cultivating HepG2 cells 

Data presentation in the results section is fine, but authors will have to improve the interpretation of experimental data in detail

Author Response

The authors have investigated and submitted a research article titled Structure and functional analysis of SHP promoter and its transcriptional response to FXRinZn-induced changes in lipid metabolism. There is a significant amount of novelty in this study that would help the current researchers who are working in this field. The current study focuses more on the journal aims. The experimental design and experimentation are fine. It may be more appropriate for publication in your journal after revision.

In this study, HepG2 cells were used instead of adipocytes since the main focus of the study was to investigate the role of Zn in modulating FXR-SHP, which plays a role in lipid metabolism.

Response: Thank you very much for your good evaluation and constructive comments. The comments are very useful for our future research. We have revised our manuscript based on your important comments, and we also listed our responses to your comments, on a point-to-point basis.

1. The authors have described several genes that are associated with lipogenesis as well as lipolysis. I would like to know whether the authors checked PPAR expression in both experimental models as well as its downstream targets.

Response: This study focused on the FXR-SHP pathway and its downstream genes, thus we did not check PPAR expression. We know that there is overlap of downstream targets between FXR-SHP and PPAR, such as cpt1α. In this study, Zn treatment increased cpt1α expression. In fact, our previous study have determined the changes of PPAR expression after Zn treatment in yellow catfish in vivo and in vitro. In that study, Zn activate multiple lipolytic pathways that include PPARα pathway (Wei et al., 2018).

Wei, C.C.; Luo, Z.; Hogstrand, C.; Xu, Y.H.; Wu, L.X.; Chen, G.H.; Pan, Y.X.; Song, Y.F. Zinc reduces hepatic lipid deposition and activates lipophagy via Zn2+/MTF‐1/PPARα and Ca2+/CaMKKβ/AMPK pathways. FASEB J. 2018, 32,

2. Please include the name of the medium used for cultivating HepG2 cells.

Response: We have listed the name of the medium (line 212-214 & 233-234). HepG2 cells were cultured in Dulbecco’s Modified Eagles Medium (DMEM) medium supplemented with 10% (v/v) fetal bovine serum (FBS).

3. Data presentation in the results section is fine, but authors will have to improve the interpretation of experimental data in detail

Response: Thank you very much for your valuable suggestions. We have added more details to interpretate the experimental data. For details, please see the Results section.

Reviewer 2 Report

The paper “Structure and functional analysis of SHP promoter and its transcriptional response to FXR in Zn-induced changes of lipid metabolism” by Gao  et al suggests a direct evidence for the interaction between FXR and SHP promoter in Pelteobagrus fulvidraco, also called Yellow catfish.  

The study is quite interesting: the introduction well introduces the study; the results are well proposed and discussed in the appropriate section. Material and methods are well described although more details could be added.  

In view of the above consideration, I suggest accepting this paper after minor revision (corrections to minor methodological errors and text editing).

Author Response

We thank the reviewer for taking the time to review our manuscript and for this positive feedback. We have added more details in Material and Methods and corrected the minor errors after double-check. Thank you again for your important comments.